# Assessing how alcohol use patterns relate to obesity among American adolescents from rural and urban areas: Five years of pooled data

Christian E. Vazquez[1]*, Fawn A. Brown[2], Faheem Ohri[1], Philip Baiden[1]

1 School of Social Work, The University of Texas at Arlington, Arlington, Texas, United States of America,
2 Department of Psychology, The University of Texas at Arlington, Arlington, Texas, United States of America

* christian.vazquez@uta.edu

**Data Availability Statement:** The data underlying the results presented in the study are available from the Substance Abuse and Mental Health Services Administration - https://www.samhsa.

## Abstract

### Background

Obesity is associated with locality and alcohol use; however, less is known about how the interaction of these two factors may compound the risk of obesity among adolescents.

### Objectives

This study examines the relationship between alcohol use and obesity among adolescents from rural and urban areas in the United States.

### Methods

Data came from a sample of American adolescents aged 12–17 years from the National Survey on Drug Use and Health (2015–2019; n = 39,489). Obesity was regressed on age, sex, race/ethnicity, income, cigarette smoking, locality, and alcohol use, with an interaction term to examine locality x alcohol use. Predicted probabilities were plotted to assess the interaction.

### Results

Compared to adolescents from urban areas, those from rural areas had 1.35 times higher odds of being obese (95% CI 1.25, 1.47). Predicted probabilities indicated that the probability of being obese was higher for rural adolescents at lower levels of drinking, up to about 40 drinks in the past 12 months.

### Conclusions

Findings suggest rural-urban differences at the intersection of alcohol use and obesity could depend on the frequency of use, but overall adolescents from rural areas may be more at risk.

gov/data/data-we-collect/nsduh-national-survey-drug-use-and-health.

**Funding:** Research reported in this publication was supported by the National Institutes of Health's National Institute of Diabetes and Digestive And Kidney Diseases of the National Institutes of Health under Award Number U24DK132740 awarded to CEV. This research was also supported by the National Institute on Aging under Award Numbers 1R24AG077012 and 5K01AG081455 awarded to CEV. The content is solely the responsibility of the authors and does not necessarily represent the official views of the National Institutes of Health.

**Competing interests:** The authors have declared that no competing interests exist.

## Introduction

Obesity affects roughly 17% of adolescents 12 to 19 years of age worldwide [1]. Another public health issue that is common among adolescents is the use of alcohol, which may exacerbate the development of obesity. Mason et al. [2] found that about one in seven adolescents aged 12 to 18 in the United States (U.S.) have engaged in alcohol use at some point in their lives. In a longitudinal study following adolescents into adulthood, it was found that heavy drinkers were more likely to be obese in young adulthood, and obese adolescents had a higher prevalence of early alcohol use when compared to adolescents who were not obese [3]. While the intersection of these two public health issues has been studied previously [4–7], few studies have examined how the intersection of these issues differs between adolescents from urban and rural localities in the U.S. A study now two decades-old found high rates of alcohol usage among adolescents both in rural and urban areas, although, those in rural areas had a higher usage and higher prevalence of being overweight or obese than adolescents living in urban or suburban areas [8]. An examination of this intersection using five years of recent nationally representative data will provide an update on the relationship between alcohol use and being obese among adolescents from rural and urban localities.

### Alcohol use

Alcohol use among adolescents in the U.S. is a major public health concern with some estimates showing that about one in six adolescents have their first alcoholic drink before age 13 [9] and one in two by age 18 [10–12]. Another study among adolescents found that most alcohol users in the sample-initiated alcohol use by age 15 [13]. The negative effects of alcohol use can last long into adulthood with literature indicating significant economic losses for drinkers compared to non-drinkers [14]. Research also suggests that a consequence of heavy drinking during adolescence is being overweight or obese in young adulthood, particularly compared to non-drinkers [15]. One explanation for the link between alcohol and obesity is the excessive amounts of calories in alcohol especially beer which has the potential to result in weight gain [16]. Overall, alcohol use has been found to be associated with obesity in young adulthood, although, less is known about how varying levels of alcohol use might be associated with obesity among adolescents [17]. Still, even less is known about how the association between alcohol use frequency and obesity among adolescents might differ by locality [18].

### Importance of place

The literature suggests there are many differences in overall health between individuals from rural and urban areas [19]. Research also indicates that rural adolescents tend to be more likely to be obese when compared to urban counterparts [20]. Further, a review study found that cross-sectional studies often find adolescents from rural areas more at risk of alcohol consumption compared to adolescents from urban areas; however, there also appear to be mixed findings [21]. Given that there is literature linking alcohol use to obesity, alcohol use to place, and obesity to place; examining how drinking, locality, and obesity are associated among adolescents is warranted. Assessing this literature, one could posit that adolescents from rural areas who drink more often might be more at risk of obesity compared to their counterparts from urban areas. Further, other evidence exists to support the idea that living in a rural area might impact factors that influence drinking or obesity. Some studies have suggested that adolescents from rural areas have more access to alcohol through their families [22] and drinking alcohol is more accepted at a younger age in rural families than in urban families [23]. Other literature suggests that people who engage in one healthy behavior, such as exercise, also engage in other healthy behaviors, such as maintaining a nutritious diet and getting sufficient

sleep, and since those from rural areas tend to exhibit worse health behaviors than their urban counterparts [24, 25], this could impact both alcohol use and obesity. Thus, findings from the alcohol and obesity literature combined with the locality findings, in addition to the lack of studies in this area comparing varying levels of drinking, justifies a deeper look into this intersection.

## The current study

The current study focuses on updating the current understanding of how the frequency of alcohol use may be associated with adolescent obesity among those from rural and urban localities with recent nationally representative data. The study adds to the obesity literature by digging deeper into the intersection of the level of alcohol use and locality (e.g., rural vs. urban). Understanding who may be most at risk of comorbidity from alcohol and obesity-related issues is important to help those implementing prevention measures. Based on the extant literature, we hypothesized that: the association between alcohol use and obesity among adolescents will differ for adolescents in rural areas from their counterparts in urban areas.

## Methods

### Sample

Data for the current study came from annual data collected by the National Survey on Drug Use and Health (NSDUH) from the years 2015–2019. Every year Substance Abuse and Mental Health Services Administration (SAMHSA) leads the NSDUH survey to collect information from the general civilian population aged 12 and older in the United States. Active-duty military, residents of institutions, and people who are homeless but not in shelters are not included in the survey population. More information about the survey methodology can be found elsewhere [26]. The information collected through these surveys includes the utilization of tobacco, alcohol, illegal drugs, substance use-related problems, psychological issues, as well as the utilization of services to treat substance abuse, and mental health issues [26]. The full sample includes 282,768 respondents. After creating a sub-sample of adolescents between the ages of 12 and 17 years who had complete data for all included variables, the analytic sample resulted in a sample of 39,489 adolescent respondents. Institutional Review Board approval was not required for this study as it is a secondary data analysis and the data are publicly available.

### Variables

**Dependent variable.** The Centers for Disease Control and Prevention (CDC) SAS code for obtaining sex and age-specific body-mass-index (BMI) percentiles was used on the dataset [27]. Being obese was determined based on the respondent being at or above the 95th percentile for children and teens of the same age and sex. This was coded into two categories, "0" as non-obese and "1" as obese.

**Independent variables.** The main independent variables were locality and alcohol use. Locality was categorized into two categories, Urban as "0" and Rural as "1". Locality was based on the 2013 Rural/Urban Continuum Codes [28], as determined by NSDUH methodology. The 2013 Rural-Urban Continuum Codes form a classification scheme that distinguishes metropolitan counties by the population size of their metro area, and nonmetropolitan counties by the degree of urbanization and adjacency to a metro area [28].

The alcohol use question used in this study asked, "What is the total number of days you drank alcohol in the past 12 months?" The response options were 1 to 365, never used, and not

used in the past 12 months. The latter two categories were re-coded into 0 = never or not in the past 365 days. This variable was treated as a continuous variable to align with previous studies using NDSUH data [29, 30].

**Control variables.** Age was a continuous variable only including adolescents aged 12 to 17 years. Sex was recoded into two categories; males coded as "0" and females coded as "1". Race/ethnicity was recoded into five categories, Whites coded as "1", Blacks coded as "2", Hispanics coded as "3", Asians coded as "4", and all Other races and mixed race coded as "5". The "Other" race category included Native American/Alaskan Native, Native Hawaiian/Other Pacific Islander, and more than one race. This group was created due to the sample size of these groups. Family income was obtained from an imputed total family income variable with four categories. Families that had annual family income less than $20,000 were coded as "1", families with annual family income from $20,000 to $49,000 were coded as "2", families with annual family income ranging from $50,000 to $74,999 were coded as "3", and all the other families that had annual family income from $75,000 or above were coded as "4". Ever smoking a cigarette was also included as control with never as "0" and ever as "1".

## Statistical analyses

The analyses were performed using SAS v. 9.3 [31] after downloading the NSDUH annual U.S. Public Use data survey files from the Substance Abuse and Mental Health Services Administration [26]. The data from the years 2015 to 2019 were pooled together. Coefficients for pooled data represent the average for the pooled years. Descriptive characteristics of the sample were examined using frequencies, percentages, means, and standard deviations (SD). Descriptive characteristics were examined for the full sample, and the rural and urban samples to highlight the rural-urban differences. The current study used binary logistic regression to examine whether the association between alcohol use frequency and obesity differs by rural versus urban locality. First, obesity was regressed on age, sex, race/ethnicity, family income, cigarette smoking, locality, and alcohol use, with an interaction term to examine locality x alcohol use. Second, predicted probabilities were plotted to assess interactions. Follow-up on the interaction was conducted even in the absence of a significant interaction at $p$ = .05 given that there is literature suggesting that the reliance of $p$ = .05 (and associated confidence intervals) for an interaction threshold is flawed because interaction tests are typically fairly low powered [32–34]. Race/ethnicity and income were dummy coded with those in the highest income category, and respondents who were White being the reference categories. To obtain odds ratios (ORs) and 95% confidence intervals (CIs), parameter estimates and their associated upper and lower confidence limits were exponentiated.

## Results

### Descriptive statistics

Table 1 shows the descriptive characteristics of the study sample. There was a higher proportion of respondents who were White within the rural sample (72%) compared to the urban sample (47%), and a higher proportion of Hispanics (26% vs 11%), Blacks (15% vs 10%), and Asians (8% vs 1%) in the urban sample compared to the rural sample. The urban sample also had a higher proportion of respondents who reported a family income of $75,000 or more compared to the rural sample (49% vs 35%). The urban sample had a higher proportion of ever smokers (91% vs 83%). The rural sample had a slightly higher average of drinking frequency (7.16 vs 5.63 days in the past 12 months). Sixteen percent of the sample was categorized as being obese, with a slightly higher proportion in the rural sample (20%) compared to the urban sample (15%). Eighty percent of the sample was from urban areas and 20% was from

**Table 1. Descriptive characteristics of the study sample, National Survey on Drug Use and Health, Years 2015–2019, Adolescent sample (age 12–17; n = 39,489).**

| | Full Sample | Rural Sample (12,255) | Urban Sample (27,234) | Chi-square/t-test |
|---|---|---|---|---|
| | n (%) or M/SD | n (%) or M/SD | n (%) or M/SD | p-value |
| Sex | | | | |
| Male | 20,319 (51.2) | 6,325 (51.8) | 13,994 (51.1) | .68 |
| Female | 19,170 (48.8) | 5,930 (48.2) | 13,240 (48.9) | |
| Age | 14.6 / 1.7 | 14.6 / 1.7 | 14.6 / 1.7 | .88 |
| 12 years | 5,486 (13.9) | 1,709 (13.7) | 3,777 (14.0) | .67 |
| 13 years | 6,313 (15.5) | 1,983 (15.9) | 4,330 (15.5) | |
| 14 years | 6,651 (17.3) | 2,006 (17.5) | 4,645 (17.3) | |
| 15 years | 6,954 (17.3) | 2,172 (17.7) | 4,782 (17.2) | |
| 16 years | 7,123 (18.1) | 2,206 (17.4) | 4,917 (18.3) | |
| 17 years | 6,692 (17.8) | 2,179 (17.9) | 4,783 (17.8) | |
| Race/ethnicity | | | | |
| White | 20,552 (51.8) | 8,628 (72.4) | 11,924 (46.7) | < .0001 |
| Black | 5,557 (14.2) | 931 (9.7) | 4,626 (15.3) | |
| Hispanic | 8,841 (23.4) | 1,389 (11.4) | 7,452 (26.4) | |
| Asian | 1,770 (6.5) | 138 (1.0) | 1,632 (7.9) | |
| Other | 2,769 (4.1) | 1,169 (5.5) | 1,600 (3.8) | |
| Family Income Level | | | | |
| Less than $20,000 | 6,109 (14.1) | 2,006 (17.2) | 4,013 (13.4) | < .0001 |
| $20,000 - $49,999 | 10,740 (25.8) | 3,773 (31.4) | 6,967 (24.4) | |
| $50,000 - $74,999 | 5,828 (13.8) | 2,127 (16.7) | 3,701 (13.0) | |
| $75,000 or More | 16,902 (46.3) | 4,349 (34.6) | 12,553 (49.2) | |
| Cigarette smoker | | | | |
| No | 34,858 (89.3) | 10,179 (83.3) | 24,679 (90.9) | < .0001 |
| Yes | 4,631 (10.7) | 2,076 (16.7) | 2,555 (9.1) | |
| Alcohol use (# of days in past 12 months) | 6.11 / 26.4 | 7.16 / 29.6 | 5.63 / 24.9 | < .0001 |
| | (range: 0–364) | (range: 0–364) | (range: 0–364) | |
| Obesity status | | | | |
| Non-Obese | 32,837 (84.0) | 9,928 (80.5) | 22,909 (85.0) | < .0001 |
| Obese | 6,652 (16.0) | 2,327 (19.5) | 4,325 (15.0) | |

Note. Percentages are based on weighted frequencies.

rural areas. S1 Table shows the same table grouped by obese vs non-obese instead of by locality.

## Logistic regression and interaction results

Logistic regression for adolescents being obese across different alcohol use patterns and locality backgrounds (see Table 2) showed that being of older age (OR = 1.05, 95% CI 1.02, 1.07), being Black (OR = 1.48, 95% CI 1.33, 1.64), being Hispanic (OR = 1.48, 95% CI 1.35, 1.63), being from the Other race group (OR = 1.22, 95% CI 1.05, 1.42), reporting an income below $20,000 (OR = 2.05, 95% CI 1.83, 2.28), between $20,000 and $49,999 (OR = 1.69, 95% CI 1.54, 1.85), or between $50,000 and $74,999 (OR = 1.47, 95% CI 1.31, 1.64), and being a cigarette ever smoker (OR = 1.30, 95% CI 1.16, 1.46) was associated with higher odds of being obese compared to being younger, White, reporting an income of $75,000 or more, or never having smoked a cigarette. Alternatively, being female (OR = 0.70, 95% CI 0.65, 0.75) and being Asian (OR = 0.54, 95% CI 0.43, 0.69) was associated with lower odds of being obese compared to

**Table 2. Logistic regression for obesity among adolescents with different alcohol use patterns x locality backgrounds (N = 39,489).**

|  | OR | 95% CI |
|---|---|---|
| Age | **1.05** | **1.02–1.07** |
| Female | **0.70** | **0.65–0.75** |
| Black | **1.48** | **1.33–1.64** |
| Hispanic | **1.48** | **1.35–1.63** |
| Asian | **0.54** | **0.43–0.69** |
| Other | **1.22** | **1.05–1.42** |
| Less than $20,000 | **2.05** | **1.83–2.28** |
| $20,000 - $49,999 | **1.69** | **1.54–1.85** |
| $50,000 - $74,999 | **1.47** | **1.31–1.64** |
| Cigarette smoker | **1.30** | **1.16–1.46** |
| Alcohol use | 1.00 | 1.00–1.00 |
| Rural | **1.35** | **1.25–1.47** |
| Alcohol use x Rural | 1.00 | 0.99–1.00 |

*Note*: **Bold** = statistically significant. OR = Odds Ratio. 95% CI = 95% Confidence Intervals. Reference groups: Male, White, $75,000 or more, Never smoker, Urban.

males and Whites. Compared to adolescents from urban areas, those from rural areas had 1.35 times higher odds of being obese (95% CI 1.25, 1.47). No statistically significant association was found for alcohol use or the alcohol use x locality interaction term.

Though the interaction term between alcohol and locality on obesity was not significant, the literature on the relationship between obesity and locality and alcohol use, and the literature on assessing interaction terms at the $p = .05$ threshold justified further investigation. Predicted probabilities were plotted to assess the interaction term (see Fig 1). The baseline (0 drinks) predicted probabilities are about 0.16 for urban adolescents and 0.19 for rural adolescents. The figure shows that there is a difference between rural and urban adolescents in the probability of being obese at the lower levels of alcohol use; however, after drinking about 50 days in the past 12 months, the difference is no longer statistically significant. Further, at about the 125 days of drinking range, the slopes cross and the predicted probability of obesity appears to become lower, ending at about 0.09, for rural adolescents and increases, ending at about 0.14, for urban adolescents; although this was not statistically significant.

## Discussion

Drawing from a nationally representative sample of adolescents in the U.S., the objective of this study was to examine the association between alcohol use frequency and obesity and whether this association differs for adolescents from urban and rural localities. We found that close to one in five adolescents were classified as obese, and the rate was higher for rural adolescents compared to urban adolescents. About 80% of the adolescents in the sample did not drink during the past 12 months. These levels were similar across rural and urban samples, with the rural sample having a slightly higher average of days drank in the past 12 months. We found partial support for our hypothesis, after controlling for other factors. Alcohol use frequency was associated with being obese differently when looking across those from rural and urban areas but was dependent on the frequency level.

The findings from the logistic regression align with previous studies suggesting being older, a male, a racial/ethnic minority (other than Asian), from a lower-income home, and being a

**Fig 1. Predicted probability plot for interaction between alcohol use and locality on adolescent obesity.** Fit computed controlling for age, sex, race/ethnicity, income, and cigarette smoking.

cigarette smoker is associated with being obese. Additionally, the literature shows consistent findings that living in a rural area is associated with being more at risk of obesity [20]. The alcohol finding should be read with caution given that while it was not statistically distinguishable in the current study, other studies have found alcohol and obesity among adolescents to be related. Thus, drinking alcohol may still be associated with being obese or, as the follow up testing suggests, there may be underlying mechanisms that are related to drinking or not drinking alcohol that may be serving a protective factor against obesity for some adolescents. Though, to be clear, we are not suggesting adolescent alcohol consumption could prevent obesity as that particular finding was not statistically significant. As past studies have found, this association may be outweighed by other factors [16, 17]. The interaction plot does suggest there is a statistically significant difference in the relationship between alcohol use frequency and being obese by locality at lower levels of alcohol use. This appears to vary by frequency level with the differences changing at higher levels. The slope of the probability also declines at a faster rate for those from rural areas, such that they start at a higher probability of obesity at lower levels of drinking and decreases by about .1 point, whereas the slope for the urban sample stays near a slope of 0. The slope for the urban sample indicates a flatter more consistent probability of obesity regardless of level of drinking. Given these findings, serial mediation analyses with variables not available in the current data (i.e., physical activity and nutrition)

may warrant consideration, given previous mixed findings associating alcohol use and adolescent obesity [17].

The relationship between alcohol use and obesity suggests a continued need to examine alcohol use and obesity association findings with different or consistent operationalizations of alcohol frequency use. Regarding the overall locality implications, it is possible that the easier access to alcohol for rural adolescents creates an environment for casual drinking which pairs with other risky behavior as suggested by the literature [22–24]. With limited studies focused on this intersection of all three factors, the current study offers insights into possible nuances based on alcohol use frequency and locality. Regarding this intersection, practitioners, and interventionists addressing obesity may get more insights from observing adolescents' relationship with alcohol that may lead to potentially risky drinking behaviors.

## Limitations

Limitations of the study include the lack of important covariates due to data restrictions. Two key variables to include in future research are physical activity and nutrition [16]. The study does pool five years of data; however, the nature of the analysis only allows for a cross-sectional snapshot of the average over the pooled years. Another limitation is that this study did not focus on co-dependence with other drugs, as the focus of the study was on alcohol use, regardless of other risk behaviors. The sample was 80% urban so the main effects may be more representative of urban adolescents than those from rural areas. There could be a potential issue with the way the alcohol frequency use variable was used (continuous vs. categories) as other forms may provide more information, however, the current study attempts to be consistent with other research assessing alcohol use frequency as a key variable [29, 30]. Future research should include an examination of co-dependence that may compound associations for adolescents from rural areas, as well as consider serial moderated mediation with key variables, such as age, race/ethnicity, income, physical activity, and nutrition.

## Conclusions

This study updates our understanding of the relationship between alcohol use frequency and being an obese adolescent from rural and urban localities in the U.S. Compared to adolescents from urban areas, adolescents from rural areas who drink at lower levels may be more at risk of having obesity, and how drinking alcohol impacts obesity at higher levels of drinking requires further examination. The intersection of alcohol use and obesity among adolescents warrants continued study given the current and mixed findings in the literature. These results offer an update for researchers to use as they assess the intersection of locality, alcohol use, and obesity.

## Supporting information

**S1 Table. Descriptive characteristics of the study sample by Obese vs Non-Obese, National Survey on Drug Use and Health, Years 2015–2019, Adolescent sample (age 12–17; n = 39,489).**
(DOCX)

## Acknowledgments

The authors would like to thank researcher Jack T. Waddell for his input on the alcohol frequency variable.

## Author Contributions

**Conceptualization:** Christian E. Vazquez, Philip Baiden.

**Data curation:** Christian E. Vazquez, Philip Baiden.

**Formal analysis:** Christian E. Vazquez, Faheem Ohri.

**Methodology:** Christian E. Vazquez, Philip Baiden.

**Software:** Christian E. Vazquez.

**Writing – original draft:** Christian E. Vazquez, Fawn A. Brown, Faheem Ohri, Philip Baiden.

**Writing – review & editing:** Christian E. Vazquez, Fawn A. Brown, Faheem Ohri, Philip Baiden.

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
