## [Decision Letter · Decision Letter 0]

19 Mar 2024

PONE-D-24-02462Assessing how alcohol use patterns impact obesity among adolescents from rural and urban areas: Five years of pooled dataPLOS ONE

Dear Dr. Vazquez,

Thank you for submitting your manuscript to PLOS ONE. After careful consideration, we feel that it has merit but does not fully meet PLOS ONE’s publication criteria as it currently stands. Therefore, we invite you to submit a revised version of the manuscript that addresses the points raised during the review process. **The revised version should address all comments. You may also note that there is a large literature on the economic losses of alcohol consumption. See ****https://doi.org/10.1002/hec.3290**

We look forward to receiving your revised manuscript.

Kind regards,

Petri Böckerman

Academic Editor

PLOS ONE

Journal Requirements:

Reviewers' comments:

Reviewer's Responses to Questions

**Comments to the Author**

1. Is the manuscript technically sound, and do the data support the conclusions?

Reviewer #1: Partly

Reviewer #2: Partly

2. Has the statistical analysis been performed appropriately and rigorously? 

Reviewer #1: No

Reviewer #2: Yes

3. Have the authors made all data underlying the findings in their manuscript fully available?

Reviewer #1: Yes

Reviewer #2: No

4. Is the manuscript presented in an intelligible fashion and written in standard English?

Reviewer #1: Yes

Reviewer #2: Yes

5. Review Comments to the Author

Reviewer #1: 1. Overall assessment

The research question here is straightforward and the method is nice and simple while sufficing for your goals, making for an unusually digestible scientific paper. The most serious problems with the analysis are the choice of subsample and the construction of not-so-simple effects, but I think both will be easy to address. See below for details.

2. Title and abstract

2.1. I don't think "Assessing how alcohol use patterns impact obesity…" is a fair title, because "impact" means "affect", and you're not looking at causal evidence. How about "relate to" instead of "impact"?

2.2. Mention that the sample comes from the US in the title or abstract. For example, you could change the title to say "among American adolescents from…".

3. Methods

3.1. "Participants who reported never using alcohol were excluded" — You should probably mention this criterion earlier, while describing the subsample of 11,814, assuming that 11,814 is what you ended up with after applying this criterion.

3.2. But the bigger issue is that the exclusion is poorly motivated. Your goal is to examine "the relationship between alcohol use and obesity". By throwing out teetotalers, you're needlessly restricting the range of your alcohol-use variable, preventing you from seeing how alcohol use is related to obesity as it increases from zero to nonzero, and you've probably made the sample a lot smaller, too. You should include the teetotalers.

3.3. Having drunk a single alcoholic beverage within the last 30 days seems like a very low bar for "frequent" drinking. That said, I recognize that your coding options are limited by the data, with this being the highest rate respondents could report, and I can't think of a better term. Maybe it's enough to describe earlier in the paper (perhaps in the abstract) what counts as "frequent" and "casual" for this study.

3.4. "The 'other' race category included Native American/Alaskan Native, Native Hawaiian/Other Pacific Islander, Asian, and more than one race. This was done due to the sample size of these groups." — I bet the result is that the "other" category is almost entirely Asian. (For example, the 2020 Census found that about 7% of the US population is Asian, while your "other" category includes 7.9% of your subsample: https://www2.census.gov/programs-surveys/decennial/2020/data/redistricting-supplementary-tables/redistricting-supplementary-table-01.pdf ) If so, it could make for somewhat more interpretable results to give Asians their own category. The size of the remaining "other" category will be very small, and hence you won't be able to infer much about it, but this avoids the situation where you attribute a specifically Asian effect to a grab bag of races.

4. Results

4.1. Self-reported health status would likely be more interpretable in its original five categories ("excellent", "very good", "good", "fair", "poor") than collapsed into two.

4.2. [In a note for Table 2] "Coefficients for controls are not shown" — Why not? They'd be easier to read in this table than where you have them right now, embedded in the prose. And the intercept is in neither the table nor the text.

4.3. In "Rural casual drinker vs. Urban frequent drinker" and "Rural frequent drinker vs. Urban casual drinker", you've varying two things at once. So, they aren't simple effects, and they don't really make sense as comparisons. Perhaps you meant to examine "Rural casual drinker vs. Rural frequent drinker" and "Urban casual drinker vs. Urban frequent drinker".

5. Discussion

5.1. "The current study findings suggest there is no difference in the association of having obesity among casual and frequent drinkers." — A failure to obtain significance is not evidence of zero association, and in fact, zero association is an implausible hypothesis. See e.g. https://stats.stackexchange.com/questions/219359

5.2. The section "Limitations" includes the sentences "This was a

large nationally representative dataset with more than sufficient sample sizes across groups." and "The current study emphasized urban versus rural differences, an understudied intersection of obesity and alcohol use." Those statements may be true, but neither is stating a limitation or is apparently relevant to the discussion of limitations.

6. Prose issues

6.1. Try to avoid circumlocutions. "Have engaged in alcohol use" is better written "have used alcohol" or "have drunk alcohol". "Being a boy" is better written "being male". "Have obesity" and similar constructions are better written "be obese" ("'0' as not having obesity" can be just "'0' as not obese"), and similarly "adolescents with obesity" is better written "obese adolescents", while "adolescent drinkers from rural areas are more at risk of being an adolescent with obesity" can be just "adolescent drinkers from rural areas are more at risk of obesity". I'd guess that the word "having" in the phrase "a consequence of heavy drinking during adolescence is overweight and having obesity" was just a typo.

6.2. "liquor" — If you mean all alcoholic beverages, it's less ambiguous to say "alcohol".

6.3. "there also appears to be mixed findings" — "appears" should be "appear".

6.4. "The data from the year 2015 to 2019" — "year" should be "years".

6.5. "compared to over $75,00" — Don't forget that last zero.

6.6. "an interaction term to examine locality x alcohol use." — Here only the phrase "alcohol use" is italicized, which is presumably a mistake.

6.7. "To obtain accurate odds ratios (ORs) and 95% confidence intervals (CIs), parameter estimates and their associated upper and lower confidence limits were exponentiated." — I'm not sure what the word "accurate" is doing here. The exponentiation isn't some kind of accuracy correction, just a rescaling transformation, from the log-odds scale to the odds scale.

Sincerely,

Kodi B. Arfer, PhD

Brown University

Reviewer #2: In the following study by Vazquez at al., the authors have attempted to analyze the relationship between being an obese adolescent and alcohol intake. Particularly, they have analyzed if there are differences between rural and urban areas and the drinking pattern: casual vs frequent drinking

Although the study is interesting from a public health perspective and the methodology sound, there are some relevant issues to be clarified. The results of the association are a little bit confusing and some important co-variables are missing.

Major comments

The question to classify subjects as casual or frequent drinkers is not direct and could lead to a misclassification. For instance, one adolescent cannot drink alcohol at all but for certain reasons could have drinked alcohol in the previous month. Then, this subject was classified as frequent drinker when in fact, was a casual one.

A more direct question, such as: “do you regularly, at least one a week or twice a month, drink alcohol?” would have been more appropriate.

This has to be acknowledged in the limitation section.

Although, I think it could be appropriate to include smoking as independent variable. I wouldn’t talk about the obesity paradox in a population of adolescents due to the fact that is related to the association of overweight or obesity, with less mortality in older people.

In the description of the logistic regression model, age appears as a risk factor along with others including black, Hispanic, Poor reported Health and low income, but the OR provided is less than 1. The authors also referred to table 2 but the detailed model with the OR for all the included variables is missing. Also in the interaction terms, it would be necessary to include the OR for each interaction category concerning to the reference category: casual drinkers of urban areas.

I am not keen on including poor reported health as an independent factor for obesity. This variable is very complex and could be significantly related with many other variables. From my point of view, probably, it is the other way around, having obesity being a risk factor for poor reported health. In this sense, I miss variables related with nutritional habits, sedentary behaviour or physical exercise. I do not know if the authors can have access or not to these variables.

The discussion seems to support that the place of living is more important than the frequency of alcohol consumption due to the fact that frequent drinkers of urban areas have less risk than casual drinkers for rural areas. So the conclusion that adolescents from rural areas who drink may be more at risk of having obesity is at least only partially supported.

Some limitations should be addressed such as the majority of the studied population belonged to urban areas as compared to rural areas. Maybe some information about the survey and the representativity of the selected population should be added.

In my opinion, it would be necessary, as it was made in table 1 for locality, a table comparing the characteristics of obese vs non-obese adolescents. Also it is interesting to highlight with an asterisk or other symbol, those differences statistically significant.

As participants who reported never using alcohol were excluded, it would be also relevant to include a supplementary table with the characteristics including also those who never reported using alcohol, and maybe comparing frequent, casual drinkers with those who have never drinked.

Minor comments

In control variables it is mentioned smoking but does not appear as a variable of adjustment in table 2.

There is a parenthesis missing in Table 1.

The introduction section is somewhat long and redundant concerning the knowledge gap and the purpose of the study.

The article does not contain figures, it would be interesting to plot the results of the interaction.

6. PLOS authors have the option to publish the peer review history of their article (what does this mean?). If published, this will include your full peer review and any attached files.

Reviewer #1: **Yes: **Kodi B. Arfer

Reviewer #2: No

---

## [Author Response · Author response to Decision Letter 0]

30 Apr 2024

Please see Response to Reviewers document attached with uploaded files.

---

## [Decision Letter · Decision Letter 1]

14 May 2024

PONE-D-24-02462R1Assessing how alcohol use patterns relate to obesity among American adolescents from rural and urban areas: Five years of pooled dataPLOS ONE

Dear Dr. Vazquez,

Thank you for submitting your manuscript to PLOS ONE. After careful consideration, we feel that it has merit but does not fully meet PLOS ONE’s publication criteria as it currently stands. Therefore, we invite you to submit a revised version of the manuscript that addresses the points raised during the review process. Please submit your revised manuscript by Jun 28 2024 11:59PM. If you will need more time than this to complete your revisions, please reply to this message or contact the journal office at plosone@plos.org. Please include the following items when submitting your revised manuscript:A rebuttal letter that responds to each point raised by the academic editor and reviewer(s). You should upload this letter as a separate file labeled 'Response to Reviewers'.A marked-up copy of your manuscript that highlights changes made to the original version. You should upload this as a separate file labeled 'Revised Manuscript with Track Changes'.An unmarked version of your revised paper without tracked changes. You should upload this as a separate file labeled 'Manuscript'.If applicable, we recommend that you deposit your laboratory protocols in protocols.io to enhance the reproducibility of your results. Protocols.io assigns your protocol its own identifier (DOI) so that it can be cited independently in the future. For instructions see: https://journals.plos.org/plosone/s/submission-guidelines#loc-laboratory-protocols. Additionally, PLOS ONE offers an option for publishing peer-reviewed Lab Protocol articles, which describe protocols hosted on protocols.io. Read more information on sharing protocols at https://plos.org/protocols?utm_medium=editorial-email&utm_source=authorletters&utm_campaign=protocols.

We look forward to receiving your revised manuscript.

Kind regards,

Petri Böckerman

Academic Editor

PLOS ONE

Journal Requirements:

**Additional Editor Comments:**

The revised version should address all remaining concerns.

Reviewers' comments:

Reviewer's Responses to Questions

**Comments to the Author**

1. If the authors have adequately addressed your comments raised in a previous round of review and you feel that this manuscript is now acceptable for publication, you may indicate that here to bypass the “Comments to the Author” section, enter your conflict of interest statement in the “Confidential to Editor” section, and submit your "Accept" recommendation.

Reviewer #2: All comments have been addressed

2. Is the manuscript technically sound, and do the data support the conclusions?

Reviewer #2: Yes

3. Has the statistical analysis been performed appropriately and rigorously? 

Reviewer #2: Yes

4. Have the authors made all data underlying the findings in their manuscript fully available?

Reviewer #2: Yes

5. Is the manuscript presented in an intelligible fashion and written in standard English?

Reviewer #2: Yes

6. Review Comments to the Author

Reviewer #2: In my view, the authors have effectively addressed all of my previous comments, resulting in improvements to the manuscript. However, I am concerned about the apparent inverse relationship observed between alcohol intake and obesity risk when alcohol intake is treated as a continuous variable. Additionally, the decision to categorize the alcohol intake variable and subsequently treat these categories as continuous is questionable. A potentially clearer approach might involve using the number of drinking days, assigning a value of 0 for those who have not consumed alcohol in the past 365 days or have never drunk alcohol.

Furthermore, in the logistic regression analysis, the odds ratio (OR) is below 1, though it is not statistically significant. Given these factors, I acknowledge that the impact of alcohol on obesity risk might differ between rural and urban areas. However, it remains unclear whether alcohol intake is a risk factor for obesity in this specific population, or conversely, if it might be protective.

Regarding the presentation of data, I recommend including the table that compares obese versus non-obese individuals as a supplementary table. Also, I am unsure if "Non-Obese" or "Not Obese," as currently used in the manuscript, is preferable. "Non-Obese" might be more standard and clearer.

7. PLOS authors have the option to publish the peer review history of their article (what does this mean?). If published, this will include your full peer review and any attached files.

Reviewer #2: No

---

## [Decision Letter · Decision Letter 2]

3 Jun 2024

Assessing how alcohol use patterns relate to obesity among American adolescents from rural and urban areas: Five years of pooled data

PONE-D-24-02462R2

Dear Dr. Vazquez,

We’re pleased to inform you that your manuscript has been judged scientifically suitable for publication and will be formally accepted for publication once it meets all outstanding technical requirements.

Kind regards,

Petri Böckerman

Academic Editor

PLOS ONE

Additional Editor Comments (optional):

I am happy with the paper. You should address the remaining technical issue(s).

Reviewers' comments:

Reviewer's Responses to Questions

**Comments to the Author**

1. If the authors have adequately addressed your comments raised in a previous round of review and you feel that this manuscript is now acceptable for publication, you may indicate that here to bypass the “Comments to the Author” section, enter your conflict of interest statement in the “Confidential to Editor” section, and submit your "Accept" recommendation.

Reviewer #2: All comments have been addressed

2. Is the manuscript technically sound, and do the data support the conclusions?

Reviewer #2: Yes

3. Has the statistical analysis been performed appropriately and rigorously? 

Reviewer #2: Yes

4. Have the authors made all data underlying the findings in their manuscript fully available?

Reviewer #2: Yes

5. Is the manuscript presented in an intelligible fashion and written in standard English?

Reviewer #2: Yes

6. Review Comments to the Author

Reviewer #2: In my view, the authors have successfully addressed all of my previous comments, resulting in notable improvements to the manuscript. The only minor suggestion I have is to simplify the legend of the figure by removing '1' and '2' and instead placing 'rural' and 'urban' directly after the descriptions of the continuous and discontinuous lines."

7. PLOS authors have the option to publish the peer review history of their article (what does this mean?). If published, this will include your full peer review and any attached files.

Reviewer #2: No

---

## [Editor Report · Acceptance letter]

18 Jun 2024

PONE-D-24-02462R2 

PLOS ONE

Dear Dr. Vazquez, 

I'm pleased to inform you that your manuscript has been deemed suitable for publication in PLOS ONE. Congratulations! Your manuscript is now being handed over to our production team.

Kind regards, 

on behalf of

Professor Petri Böckerman 

Academic Editor

PLOS ONE